# Impacts of service quality, brand image, and perceived value on outpatient's loyalty to China's private dental clinics with service satisfaction as a mediator

Wenyi Lin[1,2]*, Wanxia Yin[1]

**1** School of Public Administration, Jinan University, Guangzhou, China, **2** Common Prosperity and National Governance Institute, Jinan University, Guangzhou, China

* linwenyi2008@163.com

## Abstract

### Background

This study explores the effects and influence paths of service quality, brand image, perceived value, and service satisfaction on outpatients' loyalty to China's private dental clinics.

### Methods

A cross-sectional survey study was conducted in Dongguan City, Guangdong Province, China in January 2019. The participants were selected using the convenience sampling method. Of the 230 residents surveyed, 125 had received services in private dental clinics, being the valid sample of this study. A multiple linear regression model was used in exploring factors influencing patient loyalty. Subsequently, the path analysis was used in investigating the relationships among service quality, brand image, perceived value, patient satisfaction, and patient loyalty.

### Results

After the effects of demographic and socioeconomic variables were controlled, perceived value and patient satisfaction showed significant influences on patient loyalty. Path analysis indicated that perceived value, perceived quality, and expected quality have direct effects on patient satisfaction and have indirect effects on patient loyalty, and patient satisfaction is a mediator.

### Conclusion

Perceived service quality influences patient loyalty through the effect of patient satisfaction, which plays a key role in promoting patient loyalty. This study implies that managers in private dental clinics can gain support from customers by building customer loyalty toward dental clinics.

**Data Availability Statement:** All relevant data are within the manuscript and its Supporting information files.

**Funding:** The writing of this article was supported by Social Science Foundation of Guangdong Province (GD20CGL02). The funders had no role in the design of the study, the collection, analysis and interpretation of data.

**Competing interests:** The authors have declared that no competing interests exist.

**Abbreviations:** CCSI, Chinese Customer Satisfaction Index.

## Introduction

Patient loyalty, a behavioral intention or action to repurchase a preferred product or service consistently, is regarded as a key element of the business success of healthcare service providers [1–25]. In recent years, the Chinese government has gradually privatized healthcare services. Within this context, the private healthcare sector is growing rapidly, and the dental service industry has become a highly marketable business. The vigorous development of private dental clinics is conducive to solving the problems caused by the short supply of healthcare services in dental, general, and community hospitals. However, competition among private dental clinics is increasingly intensive [2, 3]. Hence, building patient loyalty toward dental clinics is quite important for dental service providers.

Most international empirical studies have confirmed the relationship between customer satisfaction and loyalty in the field of healthcare service [1, 4–6]. The literature about the impact of service quality on loyalty is extensive [7]. Some studies have discussed the relationships among service quality, service satisfaction, and service loyalty [8–11], whereas some analyzed the relationships among patient value, service satisfaction, and service loyalty [4, 5].

Most studies on patients' loyalty to health care institutions in China have focused on public hospitals [1, 12–21]. The context of private institutions where patients pay for services is different from the context of public healthcare service institutions where most patients get services with medical subsidies. As such, patients are more concerned with service quality in private health institutions, and private health institutions pay more attention to patients' loyalty.

Currently, only few studies in China have discussed private health service institutions [22–24]. A study considered the effect of perceived service quality on patient loyalty to private health service institutions. The results showed five perceived service quality, namely, accessibility, service attitude, medical quality, tangibility, and cost, which have a positive influence on inpatients' loyalty to private hospitals [22]. Another study that collected data in *Chongqin* municipality showed that inpatients' satisfaction plays a positive role in explaining their loyalty to hospitals [23]. A recent study conducted among 300 patients in 15 private hospitals in China indicated that service quality, patient perceived value, patient satisfaction are positively correlated with patient loyalty. Moreover, patient perceived value and patient satisfaction mediated the relationship between service quality and patient loyalty [24]. These three studies considered the impact of service quality and inpatients' satisfaction on inpatients' loyalty but did not consider the factors influencing outpatients' loyalty and the influence path of these factors on outpatients' loyalty. Considering that health services in China are undergoing privatization, this study explores the effects and influence paths of service quality, brand image, perceived value, and service satisfaction on outpatients' loyalty to China's private dental clinics. This study aims to make a contribution in the literature discussing trust-value-loyalty model (TVLM) [25] and customers in the private health service agencies. In addition, the findings of this study contribute to improving the outpatients' loyalty in China's private dental clinics.

## Methods

The research framework of this study adopted the Chinese Customer Satisfaction Index (CCSI), which was proposed by Tsinghua University in 2000. It is currently a highly authoritative and representative model in China [12–19, 22, 23]. The CCSI constructs an integrated conceptual model comprising five determinants of patient loyalty, namely, perceived value, expected value, brand image, perceived value, and patient satisfaction (Fig 1).

On the basis of a research framework, two main hypotheses were constructed.

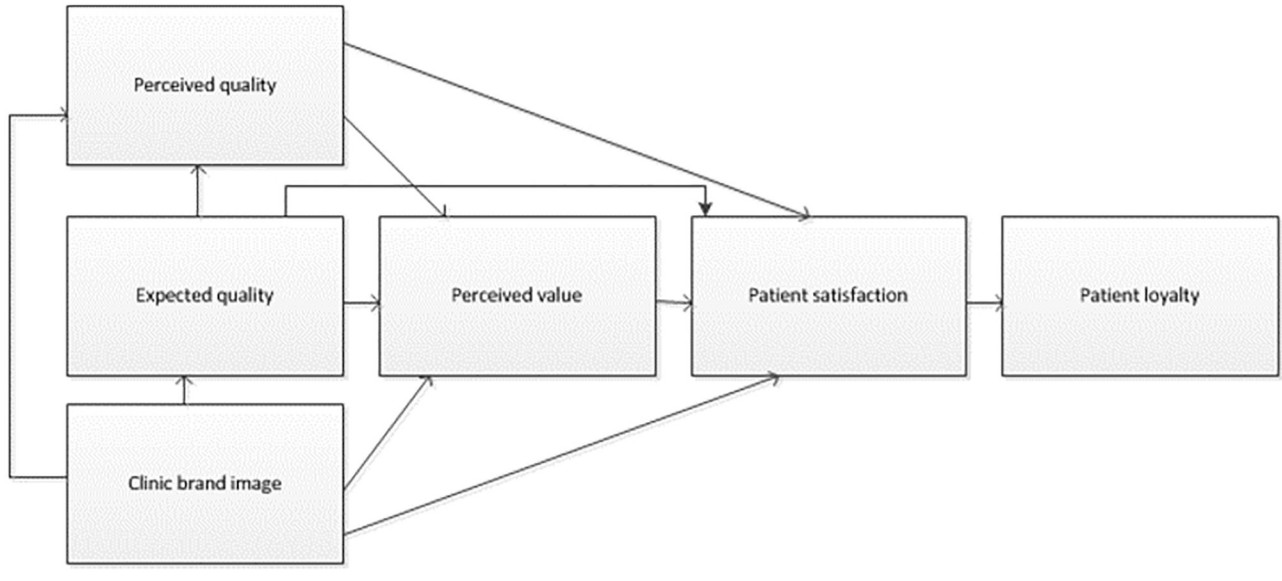

**Fig 1. Theoretical framework.**

H1: Patient satisfaction has a positive influence on patient loyalty.

H2: Perceived quality, expected quality, clinic brand image, and perceived value have indirect effects on patient loyalty, and patient satisfaction has a mediating role.

As illustrated in Table 1 regarding the patient loyalty scale, brand image refers to the clinic's recognition, credibility, and reputation. Perceived quality consists of 16 items, namely, physician medical technology, clinic medical equipment, patient treatment effect (degree of relief), handling complaints, waiting time for treatment, waiting time for payment, attitude of physician work, attitude of physician service, explanation of diagnosis and treatment, choice of treatment plan, health education, physician's clothing, sanitary conditions, environmental comfort, facility sign, and convenience service (e.g., convenience of medical transportation and distance from residence). Expected quality includes overall impression, expectations of service items before treatment, expectations of treatment outcomes, and expectations of service effect. Perceived value comprises evaluations regarding service fees compared with the quality of services received and service quality compared with service fees paid by patients. Patient satisfaction is measured by overall satisfaction, satisfaction about services provided compared with expectations, and satisfaction about private clinics compared with other types of dental clinics.

Patient loyalty includes two items: willingness to choose private clinics next time and willingness to recommend private clinics to family members or friends. Each item is coded as an ordinal variable on scales ranging from 1 to 5, and a high score indicates low satisfaction level. The Cronbach's alpha values for perceived quality, expected quality, clinic brand image, perceived value, patient satisfaction, and patient loyalty range from 0.79 to 0.97, indicating that the internal consistency of these dimensions is good.

The present study is cross-sectional survey study and conducted in Dongguan City, Guangdong Province, China in January 2019. The rationale for selecting China's private dental clinics as settings for data collection lies in the fact that dental service is the most privatized sector in China. Participants were selected with the convenience sampling method. A total of 230

**Table 1. Patient loyalty scale.**

| Brand image | Clinic's recognition |
|---|---|
| | Credibility |
| | Reputation |
| Perceived quality | Physician medical technology |
| | Clinic medical equipment |
| | Patient treatment effect (degree of relief) |
| | Handling complaints |
| | Waiting time for treatment |
| | Waiting time for payment |
| | Attitude of physician work |
| | Attitude of physician service |
| | Explanation of diagnosis and treatment |
| | Choice of treatment plan |
| | Health education |
| | Physician's clothing |
| | Sanitary conditions |
| | Environmental comfort |
| | Facility sign |
| | Convenience service |
| Expected quality | Overall impression |
| | Expectations of service items before treatment |
| | Expectations of treatment outcomes |
| | Expectations of service effects |
| Perceived value | Evaluations regarding service fees compared with the quality of services received |
| | Service quality compared with service fees paid by patients |
| Patient satisfaction | Overall satisfaction |
| | Satisfaction about services provided compared with expectations |
| | Satisfaction about private clinics compared with other types of dental clinics |
| Patient loyalty | Willingness to choose private clinics next time |
| | Willingness to recommend private clinics to family members or friends |

residents were surveyed through an online platform in March, 2019. Among them, 125 had received services in private dental clinics, being the valid sample of this study. This study obtained ethical approval from School of Public Administration, Jinan University (201909). Online informed consent was obtained from each participant who had received services from private dental clinics. At the beginning of the questionnaire, the investigator explained the purpose of the research and the confidentiality of information, and the participants who were willing to fill in the questionnaire agreed to start filling out the questionnaire online.

Descriptive statistics were used in analyzing a sample's characteristics. Then a multiple linear regression model was used in exploring the effects of service quality, brand image, perceived value, and patient satisfaction on patient loyalty. Unstandardized coefficient (B), standardized coefficient (Beta), and 95% confidence intervals (CIs) were reported, and the $P$ values reported were two-tailed. The influences of service quality, brand image, perceived value, and patient satisfaction on patient loyalty were explored through path analysis. The maximum-likelihood estimation method was used in path analysis. The $\chi^2/df$, RMSEA, CFI, and SRMR indices were used in evaluating the fit of the analytical model. All data were

analyzed with SPSS 24.0 and AMOS 21.0 (International Business Machines Corp: Beijing, China). Respondents with one missing variable were excluded from analysis.

## Results

Table 2 reports the characteristics of the respondents. Among the 125 respondents, 44 were men, accounting for 35.20% of the total sample. The majority of the participants were under 45 years old. The number of respondents who completed undergraduate or junior college was the highest, accounting for 64% of the total sample. Most participants (39.20% in total) had incomes of 1720 RMB and below.

In 125 respondents, the mean of patient loyalty was 7.38 (SD = 1.57) out of 10. The average satisfaction score of all respondents was 11.08 (SD = 2.05) out of 15. The mean perceived value was 7.14 (SD = 1.43) out of 10. The mean of the clinic brand image was 10.98 (SD = 1.99) out of 15. The mean of expected quality was 14.92 (SD = 2.43) out of 20. The average score of perceived quality was 60.01 (SD = 10.00) out of 80.

The effects of perceived quality, expected quality, clinic brand image, perceived value, and patient satisfaction on patient loyalty are shown in Table 3. After the effects of demographic and socioeconomic variables were controlled, perceived value (Beta = 0.249; 95% CI: 0.118–0.380) and patient satisfaction (Beta = 0.372; 95% CI: 0.175–0.570) had positive influences on patient loyalty. By contrast, perceived quality, expected quality, and clinic brand image were not significantly associated with patient loyalty.

**Table 2. Sample description (n = 125).**

|  | Mean or percent | Standard deviation | Minimum | Maximum |
|---|---|---|---|---|
| Patient loyalty | 7.38 | 1.57 | 2 | 10 |
| Patient satisfaction | 11.08 | 2.05 | 3 | 15 |
| Perceived value | 7.14 | 1.43 | 4 | 10 |
| Clinic brand image | 10.98 | 1.99 | 3 | 15 |
| Expected quality | 14.92 | 2.43 | 8 | 20 |
| Perceived quality | 60.01 | 10.00 | 16 | 80 |
| Gender(male) | 35.20% |  |  |  |
| Age |  |  |  |  |
| 25 years old and below | 56.80% |  |  |  |
| 26–45 years old | 21.60% |  |  |  |
| 46–60 years old | 13.60% |  |  |  |
| 61 years old and above | 8.00% |  |  |  |
| Education |  |  |  |  |
| Elementary school and below | 9.60% |  |  |  |
| Middle school | 13.60% |  |  |  |
| High school or vocational school | 10.40% |  |  |  |
| College | 64.80% |  |  |  |
| Graduate school | 1.60% |  |  |  |
| Income per month |  |  |  |  |
| 1720 RMB and below | 39.20% |  |  |  |
| 1721–4000 RMB | 28.80% |  |  |  |
| 4001–6000 RMB | 16.80% |  |  |  |
| 6001–8000 RMB | 10.40% |  |  |  |
| 8001 RMB and above | 4.80% |  |  |  |

**Table 3. Multiple linear regression model of patient loyalty in the private dental clinics (n = 125).**

|  | B (95% CI*) | Beta (95% CI*) | P-value |
|---|---|---|---|
| Constant | -0.445 (-2.103–1.213) |  | 0.596 |
| Perceived quality | 0.023 (-0.010–0.056) | 0.148(-0.063–0.360) | 0.168 |
| Expected quality | 0.006 (-0.104–0.116) | 0.009(-0.161–0.180) | 0.914 |
| Clinic brand image | 0.126 (-0.017–0.269) | 0.160(-0.022–0.342) | 0.084 |
| Perceived value | 0.274 (0.130–0.418) | 0.249(0.118–0.380) | <0.001 |
| Patient satisfaction | 0.284 (0.134–0.435) | 0.372(0.175–0.570) | <0.001 |
| Gender | -0.120 (-0.470–0.230) | -0.037(-0.144–0.070) | 0.497 |
| Age | 0.077 (-0.178–0.331) | 0.048(-0.111–0.206) | 0.553 |
| Education | -0.074 (-0.308–0.160) | -0.050(-0.208–0.108) | 0.533 |
| Monthly income | 0.076 (-0.062–0.214) | 0.057(-0.047–0.162) | 0.279 |

*CI- confidence interval.

The path influence of perceived quality, expected quality, brand image, perceived value, and patient satisfaction on patient loyalty were analyzed through path analysis, as shown in Table 4. Perceived value and patient satisfaction influenced patient loyalty significantly and directly. Perceived value, perceived quality, and expected quality had indirect effects on patient loyalty, and patient satisfaction was a mediator.

## Discussion

This study explored the effects and influence path of service quality, brand image, perceived value, and service satisfaction on outpatients' loyalty in China's private dental clinics. The findings indicated that adjusted demographic and socioeconomic variables, perceived value, and patient satisfaction have positive and direct influences on patient loyalty. Additionally, perceived value, perceived quality, and expected quality influenced patient loyalty by affecting patient satisfaction.

A positive impacts of outpatients' perceived quality, perceived value, and service satisfactory on patients' loyalty in public health institutions were explored in previous studies [1, 14, 17–19]. The overall image of a hospital and perceived quality in terms of technical service levels of

**Table 4. The path coefficients in SEM (n = 125).**

| Variables |  | Variables | Estimate (Unstandarized) | Estimate (Standardized) | S.E. | C.R. | P |
|---|---|---|---|---|---|---|---|
| Expected quality | < --- | Brand image | 0.933 | 0.764 | .071 | 13.174 | <0.001 |
| Perceived quality | < --- | Expected quality | 1.675 | 0.408 | .345 | 4.859 | <0.001 |
| Perceived quality | < --- | Brand image | 2.213 | 0.441 | .421 | 5.255 | <0.001 |
| Perceived value | < --- | Brand image | 0.053 | 0.074 | .085 | .625 | .532 |
| Perceived value | < --- | Expected quality | -0.007 | -0.012 | .069 | -.102 | .919 |
| Perceived value | < --- | Perceived quality | 0.084 | 0.589 | .016 | 5.139 | <0.001 |
| Patient satisfaction | < --- | Perceived value | 0.196 | 0.136 | .083 | 2.353 | .019 |
| Patient satisfaction | < --- | Brand image | 0.091 | 0.088 | .079 | 1.156 | .248 |
| Patient satisfaction | < --- | Perceived quality | 0.113 | 0.551 | .017 | 6.774 | <0.001 |
| Patient satisfaction | < --- | Expected quality | 0.160 | 0.189 | .064 | 2.510 | .012 |
| Patient loyalty | < --- | Patient satisfaction | 0.465 | 0.440 | .050 | 9.270 | <0.001 |
| Patient loyalty | < --- | Perceived value | 0.317 | 0.289 | 0.072 | 4.398 | <0.001 |

Model fitness: $\chi 2/df$ 4.687(P <0.05); RMSEA: 0.172 (90% CI: 0.089–0.268); CFI: 0.982; TLI: 0.909, SRMR: 0.000.

doctors [13, 15, 17, 18], clinic's facilities and environment [15, 19], waiting time [13], and doctor's attitude [16, 17, 22] influence inpatients' loyalty to doctors. Patient satisfaction is a mediator between perceived value and patient loyalty [12]. Our findings confirmed these relationships in private dental clinics. This study is the first time to examine CCSI model in private dental clinics in China.

This study is limited to private dental clinics, and the convenience sampling is applied to sample selection. Thus, the findings may not be generalized to all dental clinics due to selection bias. Moreover, the sample of this study cannot represent the population in China. Studies with larger samples are needed for increasing understanding of the relationships among service quality, satisfaction, and loyalty. Future studies can replicate this study design in other private dental clinics, to enhance the generalizability of the findings. Nevertheless, this study can help managers in private dental clinics in China in gaining support from customers through building customer loyalty toward dental services.

Furthermore, the findings of this study provide some practical implications. In practice, several dimensions associated with perceived service quality and perceived value should be improved in private clinics. Perceived service quality in this study included two items: waiting time for treatment and convenience service (e.g., convenience of medical transportation and distance from residence). The perceived value comprised the evaluations regarding service fees compared with the quality of services received and service quality compared with the service fee by patients.

The first concern is appointment service. Most private dental clinics do not have online appointment services in China, and thus potential patients cannot make appointments in advance and waiting time increases. When waiting time is extremely long, patients are less likely to receive services [15, 20, 26].

The second concern is the location of private dental clinics. Currently, high-quality large-scale private dental clinics are usually located in urban centers and are separated from residential communities, thereby reducing convenience. Popa and Daniela [27] and Crutzen et al. [28] claimed that online tools can provide convenient services for patients and can enhance patient's loyalty to hospitals. On this basis, private dental clinics can fully use a *WeChat* public number or *WeChat* small program to provide a range of services, such as medical consultation, appointment registration, online diagnosis, fee inquiry, and online payment.

Another concern is service fee. When the investigators conducted the survey, most private dental clinics in Dongguan City do not have a clear price list and charging standards, which are generally in the form of fees for doctors' oral quotations, doctors paying bills, and no billing information. This type of charging method may easily lead to random pricing and disorder within the dental clinics, which may reduce the satisfaction of perceived value.

To standardize the development of dental clinics, health authorities should strengthen the supervision of private dental clinics, and daily supervision and inspection of business qualifications, service quality, equipment safety, and price expenses should be strengthened. Moreover, examining and determining the qualifications of dentists are necessary [29].

## Supporting information

**S1 File. Questionnaire, Chinese version.**
(DOCX)

**S2 File. Questionnaire, English translation.**
(DOCX)

**S3 File. Data.**
(SAV)

## Author Contributions

**Conceptualization:** Wenyi Lin.

**Formal analysis:** Wenyi Lin.

**Investigation:** Wanxia Yin.

**Methodology:** Wenyi Lin.

**Writing – original draft:** Wenyi Lin.

**Writing – review & editing:** Wenyi Lin.

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
