## [Decision Letter · Decision Letter 0]

4 Apr 2022

PONE-D-21-24436

Impacts of service quality, brand image, and perceived value on outpatient’s loyalty to China’s private dental clinics with service satisfaction as a mediator

PLOS ONE

Dear Dr. Lin,

Thank you for submitting your manuscript to PLOS ONE. After careful consideration, we feel that it has merit but does not fully meet PLOS ONE’s publication criteria as it currently stands. Therefore, we invite you to submit a revised version of the manuscript that addresses the points raised during the review process.

We look forward to receiving your revised manuscript.

Kind regards,

Bing Xue, Ph.D.

Academic Editor

PLOS ONE

a) Did participants provide their written or verbal informed consent to participate in this study?

5. Thank you for stating the following in the Funding Section of your manuscript:

“The writing of this article was supported by Social Science Foundation of Guangdong Province（GD20CGL02）. The funders had no role in the design of the study, the collection, analysis and interpretation of data.”

“The writing of this article was supported by Social Science Foundation of Guangdong Province（GD20CGL02）. The funders had no role in the design of the study, the collection, analysis and interpretation of data.”

6. Please include your tables as part of your main manuscript and remove the individual files. Please note that supplementary tables (should remain/ be uploaded) as separate ""supporting information"" files.

Reviewers' comments:

Reviewer's Responses to Questions

**Comments to the Author**

1. Is the manuscript technically sound, and do the data support the conclusions?

Reviewer #1: Yes

Reviewer #2: Yes

Reviewer #3: Yes

Reviewer #4: Yes

Reviewer #5: Yes

2. Has the statistical analysis been performed appropriately and rigorously? 

Reviewer #1: Yes

Reviewer #2: Yes

Reviewer #3: Yes

Reviewer #4: Yes

Reviewer #5: Yes

3. Have the authors made all data underlying the findings in their manuscript fully available?

Reviewer #1: Yes

Reviewer #2: Yes

Reviewer #3: Yes

Reviewer #4: Yes

Reviewer #5: Yes

4. Is the manuscript presented in an intelligible fashion and written in standard English?

Reviewer #1: Yes

Reviewer #2: Yes

Reviewer #3: Yes

Reviewer #4: No

Reviewer #5: Yes

5. Review Comments to the Author

Reviewer #1: Thank you for providing me the opportunity to read an interesting article. Although the idea is unique and exciting, a few areas need to improve before publication. The contributions of the article need more emphasis. The literature regarding the study is not up to date. The authors are advised to add some recent literature from 2021 and 2022. Good Luck

Reviewer #2: I am glad to review and assess this exciting article, entitled " Impacts of service quality, brand image, and perceived value on outpatient’s loyalty to China’s private dental clinics with service satisfaction as a mediator" . I am fully satisfied and accept this manuscript for publication

Reviewer #3: Great work. It makes a very insightful reading . The analysis are very well done and the recommendation is apt. A good demonstration of the principles of research. Looking forward for more research and update with same field .

Reviewer #4: This research focuses on the impacts and influence paths of service quality, perceived value, brand image, and service satisfaction on outpatients’ loyalty to China’s private dental clinics.

This paper's contention is built on a suitable base of hypotheses, concepts, or other thoughts and is also well designed with appropriate methods.

Findings displayed clearly and examined suitably

This paper identify clearly between any implications for research but need to mention details about theoretical implications and Practical implications. This paper have bridge the gap between theory and practice.

The authors need to recheck the clarity of expression and readability, such as sentence structure.

1. Page 4, line 4: Need to cite more references from the previous study.

2. Contributions of the paper is convincing.

3. page 9, second paragraph need more citation.

4. It requests professional editing so that readers' coherence can be improved.

Reviewer #5: In China, where private health care services are very expansive, the study with this framework is interesting. Overall, this study is presented well. It would be good to work more on the limitation section.

6. PLOS authors have the option to publish the peer review history of their article (what does this mean?). If published, this will include your full peer review and any attached files.

Reviewer #1: No

Reviewer #2: No

Reviewer #3: **Yes: **Haitham Medhat Abdelaziz Aboulilah

Reviewer #4: No

Reviewer #5: No

---

## [Author Response · Author response to Decision Letter 0]

28 Apr 2022

Responses to Journal requirements:

Response: We have carefully read PLOS ONE's style requirements, and revised the manuscript’s format based on the requirements.

a) Did participants provide their written or verbal informed consent to participate in this study?

Response: Verbal consent was obtained from each participant. 

Response: Written informed consent cannot be obtained because the investigators collected survey data through the online platform in which getting written verification from the participant is not practical, please see Page 5, Line10-12. We sent a survey link in different online groups, and stated the purpose of this study clearly, and then asked participants whether they were likely to fill in the survey, if they approved, they could finished the survey online. This study got ethical approval from the Ethics Committee of School of Public Administration, Jinan University. The Ethics Committee of School of Public Administration, Jinan University approved the consent procedure. 

Response: Yes. In the revised version, we have provided the questionnaire as part of this study. Please the supporting files.

Response: Thanks. We will make a revision when resubmitting the manuscript. 

5. Thank you for stating the following in the Funding Section of your manuscript:

“The writing of this article was supported by Social Science Foundation of Guangdong Province（GD20CGL02）. The funders had no role in the design of the study, the collection, analysis and interpretation of data.”

“The writing of this article was supported by Social Science Foundation of Guangdong Province（GD20CGL02）. The funders had no role in the design of the study, the collection, analysis and interpretation of data.”

Response: Thanks. We have removed the funding information in the revised version and plan to provide the funding information in the Funding Statement section of the online submission form.

6. Please include your tables as part of your main manuscript and remove the individual files. Please note that supplementary tables (should remain/ be uploaded) as separate ""supporting information"" files.

Response: Thanks. We have inserted the figure and tables in the main manuscript. 

Response: Base on the reviewers’ suggestions, we have added the recent literature in the revised version. Concretely, the updating literature included: 

6. Nguyen NX, Tran K, Nguyen TA. Impact of Service Quality on In-Patients' Satisfaction, Perceived Value, and Customer Loyalty: A Mixed-Methods Study from a Developing Country. Patient Prefer Adherence. 2021;15:2523-2538. Published 2021 Nov 17. doi:10.2147/PPA.S333586.

20. Liu S, Li G, Liu N, Hongwei W. The Impact of Patient Satisfaction on Patient Loyalty with the Mediating Effect of Patient Trust. Inquiry.

2021;58:469580211007221. doi:10.1177/00469580211007221.

21. Zhang L, Zhang Q, Li X, et al. The effect of patient perceived involvement on patient loyalty in primary care: The mediating role of patient satisfaction and the moderating role of the family doctor contract service. Int J Health Plann Manage. 2022;37(2):734-754. doi:10.1002/hpm.3355.

24. Guo Y, Zhou Y, Xing X, Li X. Exploring the Relationship between Service Quality of Private Hospitals and Patient Loyalty from the Perspective of Health Service. Iran J Public Health. 2020;49(6):1097-1105.

25. Gidaković, Petar and Vesna Žabkar. “How industry and occupational stereotypes shape consumers' trust, value and loyalty judgments concerning service brands.” Journal of Service Management 2021; 6:92-113.

28. Bielen, Frédéric & Demoulin, Nathalie. Waiting time influence on the satisfaction-loyalty relationship in services. Managing Service Quality. 2007; 17. 174-193. 10.1108/09604520710735182.

29. Bezerra de Oliveira, Lucas Ambrósio et al. “What determines patient loyalty in health services? An analysis to assist service quality management.” Total Quality Management & Business Excellence.2021; DOI: 10.1080/14783363.2021.1960500. 

Responses to Reviewers' comments:

Reviewer's Responses to Questions

Reviewer #1: Thank you for providing me the opportunity to read an interesting article. Although the idea is unique and exciting, a few areas need to improve before publication. The contributions of the article need more emphasis. The literature regarding the study is not up to date. The authors are advised to add some recent literature from 2021 and 2022. Good Luck.

Response: Thanks. We have stated contributions of the article more clearly in the revised manuscript, please see Page 3 Lines11-14:” This study aims to make a contribution in the literature discussing trust-value-loyalty model（TVLM）[25] and customers in the private health service agencies. In addition, the findings of this study contribute to improving the outpatients’ loyalty in China’s private dental clinics.” . And we have provided updated literature in the revised manuscript, please see Page 3 Line1-5: “A recent study conducted among 300 patients in 15 private hospitals in China indicated that service quality, patient perceived value, patient satisfaction are positively correlated with patient loyalty. Moreover, patient perceived value and patient satisfaction mediated the relationship between service quality and patient loyalty [24]. “Page 3 Line11-12 “T This study aims to make a contribution in the literature discussing trust-value-loyalty model（TVLM）[25]”. We have searched the latest articles published on the journals indexed in the web of science in theses two years, but the articles related to the customers’ loyalty in the private health service agencies are scarce. 

Reviewer #2: I am glad to review and assess this exciting article, entitled " Impacts of service quality, brand image, and perceived value on outpatient’s loyalty to China’s private dental clinics with service satisfaction as a mediator" . I am fully satisfied and accept this manuscript for publication

Reviewer #3: Great work. It makes a very insightful reading . The analysis are very well done and the recommendation is apt. A good demonstration of the principles of research. Looking forward for more research and update with same field .

Reviewer #4: This research focuses on the impacts and influence paths of service quality, perceived value, brand image, and service satisfaction on outpatients’ loyalty to China’s private dental clinics.

This paper's contention is built on a suitable base of hypotheses, concepts, or other thoughts and is also well designed with appropriate methods.

Findings displayed clearly and examined suitably

This paper identify clearly between any implications for research but need to mention details about theoretical implications and Practical implications. This paper have bridge the gap between theory and practice.

Response: Thanks. We have stated more theoretical contributions and Practical implications on Page 3 Lines 11-14 :” This study aims to make a contribution in the literature discussing trust-value-loyalty model（TVLM）[25] and customers in the private health service agencies. In addition, the findings of this study contribute to improving the outpatients’ loyalty in China’s private dental clinics.”

The authors need to recheck the clarity of expression and readability, such as sentence structure.

Response: We have asked native speaker to proofread the revised manuscript, and tried our best to make the language more readable.

1.: Need to cite more references from the previous study.

Response: Thanks. We have cited more references from the previous studies, please see references:

6. Nguyen NX, Tran K, Nguyen TA. Impact of Service Quality on In-Patients' Satisfaction, Perceived Value, and Customer Loyalty: A Mixed-Methods Study from a Developing Country. Patient Prefer Adherence. 2021;15:2523-2538. Published 2021 Nov 17. doi:10.2147/PPA.S333586.

20. Liu S, Li G, Liu N, Hongwei W. The Impact of Patient Satisfaction on Patient Loyalty with the Mediating Effect of Patient Trust. Inquiry.

2021;58:469580211007221. doi:10.1177/00469580211007221.

21. Zhang L, Zhang Q, Li X, et al. The effect of patient perceived involvement on patient loyalty in primary care: The mediating role of patient satisfaction and the moderating role of the family doctor contract service. Int J Health Plann Manage. 2022;37(2):734-754. doi:10.1002/hpm.3355.

24. Guo Y, Zhou Y, Xing X, Li X. Exploring the Relationship between Service Quality of Private Hospitals and Patient Loyalty from the Perspective of Health Service. Iran J Public Health. 2020;49(6):1097-1105.

25. Gidaković, Petar and Vesna Žabkar. “How industry and occupational stereotypes shape consumers' trust, value and loyalty judgments concerning service brands.” Journal of Service Management 2021; 6:92-113.

28. Bielen, Frédéric & Demoulin, Nathalie. Waiting time influence on the satisfaction-loyalty relationship in services. Managing Service Quality. 2007; 17. 174-193. 10.1108/09604520710735182.

29. Bezerra de Oliveira, Lucas Ambrósio et al. “What determines patient loyalty in health services? An analysis to assist service quality management.” Total Quality Management & Business Excellence.2021; DOI: 10.1080/14783363.2021.1960500. 

2. Contributions of the paper is convincing.

3. page 9, second paragraph need more citation.

Response: Thanks. We have cited more references from the previous studies, please see Page 10 Line12 “[15,20,26]”.

4. It requests professional editing so that readers' coherence can be improved.

Response: We have asked native speaker to proofread the revised manuscript, and tried our best to make the language more readable.

Reviewer #5: In China, where private health care services are very expansive, the study with this framework is interesting. Overall, this study is presented well. It would be good to work more on the limitation section.

Response: Thanks. We have stated more limitations on Page9,Lines 20-27 and Page 10, Line 1: “This study is limited to private dental clinics, and the convenience sampling is applied to sample selection. Thus, the findings may not be generalized to all dental clinics due to selection bias. Moreover, the sample of this study cannot represent the population in China. Studies with larger samples are needed for increasing understanding of the relationships among service quality, satisfaction, and loyalty. Future studies can replicate this study design in other private dental clinics, to enhance the generalizability of the findings. Nevertheless, this study can help managers in private dental clinics in China in gaining support from customers through building customer loyalty toward dental services. ”

---

## [Editor Report · Decision Letter 1]

18 May 2022

Impacts of service quality, brand image, and perceived value on outpatient’s loyalty to China’s private dental clinics with service satisfaction as a mediator

PONE-D-21-24436R1

Dear Dr. Lin,

We’re pleased to inform you that your manuscript has been judged scientifically suitable for publication and will be formally accepted for publication once it meets all outstanding technical requirements.

Kind regards,

Bing Xue, Ph.D.

Academic Editor

PLOS ONE
---

## [Editor Report · Acceptance letter]

24 May 2022

PONE-D-21-24436R1 

Impacts of service quality, brand image, and perceived value on outpatient’s loyalty to China’s private dental clinics with service satisfaction as a mediator 

Dear Dr. Lin:

I'm pleased to inform you that your manuscript has been deemed suitable for publication in PLOS ONE. Congratulations! Your manuscript is now with our production department. 

Kind regards, 

on behalf of

Professor Bing Xue 

Academic Editor

PLOS ONE